# A Multicomponent Protocol for the Synthesis of Highly Functionalized γ-Lactam Derivatives and Their Applications as Antiproliferative Agents

**DOI:** 10.3390/ph14080782

**Published:** 2021-08-09

**Authors:** Xabier del Corte, Adrián López-Francés, Aitor Maestro, Ilia Villate-Beitia, Myriam Sainz-Ramos, Edorta Martínez de Marigorta, José Luis Pedraz, Francisco Palacios, Javier Vicario

**Affiliations:** 1Departamento de Química Orgánica I, Centro de Investigación y Estudios Avanzados “Lucio Lascaray”-Facultad de Farmacia, University of the Basque Country (UPV/EHU), Paseo de la Universidad 7, 01006 Vitoria-Gasteiz, Spain; xabier.delcorte@ehu.eus (X.d.C.); adrian.lopez@ehu.eus (A.L.-F.); aitor.maestro@ehu.eus (A.M.); edorta.martinezdemarigorta@ehu.eus (E.M.d.M.); 2NanoBioCel Group, Facultad de Farmacia, University of the Basque Country (UPV/EHU), 01006 Vitoria-Gasteiz, Spain; aneilia.villate@ehu.eus (I.V.-B.); miriam.sainz@ehu.eus (M.S.-R.); 3Biomedical Research Networking Center in Bioengineering, Biomaterials and Nanomedicine (CIBER-BBN), Facultad de Farmacia, University of the Basque Country (UPV/EHU), 01006 Vitoria-Gasteiz, Spain; 4Bioaraba, NanoBioCel Research Group, Facultad de Farmacia, University of the Basque Country (UPV/EHU), 01006 Vitoria-Gasteiz, Spain

**Keywords:** γ-lactams, multicomponent synthesis, antiproliferative effect

## Abstract

An efficient synthetic methodology for the preparation of 3-amino 1,5-dihydro-2*H*-pyrrol-2-ones through a multicomponent reaction of amines, aldehydes, and pyruvate derivatives is reported. In addition, the densely substituted lactam substrates show in vitro cytotoxicity, inhibiting the growth of carcinoma human tumor cell lines HEK293 (human embryonic kidney), MCF7 (human breast adenocarcinoma), HTB81 (human prostate carcinoma), HeLa (human epithelioid cervix carcinoma), RKO (human colon epithelial carcinoma), SKOV3 (human ovarian carcinoma), and A549 (carcinomic human alveolar basal epithelial cell). Given the possibilities in the diversity of the substituents that offer the multicomponent synthetic methodology, an extensive structure-activity profile is presented. In addition, both enantiomers of phosphonate-derived γ-lactam have been synthesized and isolated and a study of the cytotoxic activity of the racemic substrate vs. its two enantiomers is also presented. Cell morphology analysis and flow cytometry assays indicate that the main pathway by which our compounds induce cytotoxicity is based on the activation of the intracellular apoptotic mechanism.

## 1. Introduction

Population aging is one of humanity’s greatest achievements, but also one of its biggest challenges. From a health point of view, one of the main side effects of the growth in life expectancy is the increasing occurrence of disorders associated with aging. In this regard, cancer diseases are currently one of the biggest health problems worldwide [1] and, accordingly, malignant neoplasms have become nowadays one of the primary targets in medicinal sciences. One of our strongest weapons to combat cancer implies the combination of surgical techniques and chemotherapeutical agents, where antineoplastic drugs are used to kill fast-growing cells in the body [2]. For many years, one of the major goals of cancer therapy has been the specific and effective elimination of tumor cells by apoptosis, a programmed and highly regulated cellular death mechanism triggered by multiple factors that induce cellular stress. In this regard, there are already available some FDA-approved anticancer agents specifically designed to target intracellular apoptotic pathways, and also novel more selective agents are under development [3]. During the last decades, a plethora of cytotoxic drugs that target actively proliferating cells have been developed, but more efforts are still required in this area, to increase our ammunition to fight against cancer. The identification of new active compounds is an imperative task in medicine and, there, organic and medicinal chemistry plays a crucial role, as the front line of battle, through the production of libraries of organic compounds and the preliminary in vitro evaluation of their cytotoxicity against cancer cell lines.

In this regard, the γ-lactam ring (Figure 1) is a key structural scaffold in medicinal chemistry that can be found in the structure of many natural and synthetic bioactive compounds [4]. Within this family of heterocycles, 1,5-dihydro-2*H*-pyrrol-2-ones (Figure 1), present a conjugated ring system, that possesses latent reactivity for further modifications and, for this reason, they are valuable building blocks in synthetic chemistry [5,6,7,8]. Moreover, the structure of these unsaturated γ-lactam derivatives is the essential part of the skeleton of numerous relevant molecules that show a large variety of biological activities [9,10,11]. For example, ascosali pyrrolidone A has been isolated from the marine fungus *Ascochyta salicornae* and shows antimicrobial activity [12], cryptocin is inactive against human fungi but shows selective activity against plant pathogenic ones [13] and erisothramidine 2 shows anticonvulsive, sedative, and hypnotic properties [14] and has been reported as a neuromuscular blocking agent [15] (Figure 1).

Interestingly, some 2-pyrrolone substrates have been recently identified as p53−MDM2 [16] and STAT3 [17] inhibitors, which results in a strong antiproliferative activity and, besides, many other 2-pyrrolone derivatives have been described as antitumoral agents [18,19,20,21,22,23].

One of the most widely used methodologies for the construction of the skeleton of 1,5-dihydro-2*H*-pyrrol-2-ones consists of the three-component reaction of amines with aldehydes and pyruvate derivatives in the presence of an acid catalyst, that leads to the formation of 3-amino substituted 1,5-dihydro-2*H*-pyrrol-2-ones [24,25,26,27]. Alternatively, acetylene carboxylates can be used instead of pyruvate derivatives [28,29]. 3-Amino 1,5-dihydro-2*H*-pyrrol-2-ones are indeed cyclic α-dehydro α-amino acids and such skeleton is known to be present in many bioactive molecules as antimicrobials with anti-biofilm activity, caspase-3 inhibitors, analgesics, or antipyretics [30,31,32,33,34] and it is also the basic structure of dithiopyrrolone antibiotics [35].

Besides, multicomponent protocols are considered as an essential tool in diversity-oriented synthesis [36,37], due to the high degree of molecular diversity achieved, and, accordingly, they have become a preferential methodology in the field of medicinal chemistry [38,39]. In this context, a few years ago, we described a sulfuric acid-promoted multicomponent reaction of amines, aldehydes, and ethyl pyruvate to efficiently afford 3-amino 1,5-dihydro-2*H*-pyrrol-2-ones [24]. More recently, we have reported the asymmetric version of this reaction, using chiral phosphoric acids as catalysts [27], and we have extended this methodology to the synthesis of phosphorated and fluorinated γ-lactam derivatives [40]. We have also described a similar Brönsted acid-catalyzed multicomponent reaction using acetylene carboxylates instead of pyruvate derivatives that affords in this case 4-carboxyl substituted γ-lactam derivatives [29].

Likewise, we have been recently involved in the development of strategies for the preparation of fluorine [41,42,43,44] and phosphorus [45,46,47,48,49] containing derivatives, as well as for the design of new biologically active simple and fused heterocycles with anticancer and antileishmanial activity [50,51,52,53,54,55,56]. Due to the potential of five-membered heterocycles containing the 1,5-dihydro-2*H*-pyrrol-2-one skeleton as anticancer agents, we thought that γ-lactam derivatives, obtained by multicomponent methodologies, may be excellent candidates for the development of new antiproliferative agents.

## 2. Results and Discussion

### 2.1. Chemistry

The multicomponent protocol for the synthesis of 1,5-dihydro-2*H*-pyrrol-2-ones **4**–**12** implies the reaction of aromatic amines **1** with aldehydes **2** and pyruvate derivatives **3** in the presence of a catalytic amount of BINOL derived phosphoric for several hours, at room temperature [27,40]. Although the stoichiometry of the reaction is indeed 2:1:1 (amine/aldehyde/pyruvate), three equivalents of pyruvate derivative are required to achieve good yields in reasonable reaction times. Due to the good solubility of the reagents and phosphoric acid catalyst in chlorinated solvents, dichloromethane was chosen as the optimal reaction media, since the reaction in ethereal solvents, toluene, or AcN afforded γ-lactam derivatives in lower yields and higher reaction times. In addition, the presence of MgSO_4_ is necessary to remove the water generated in the process (Scheme 1). Although the reaction proceeds with similar efficiency using other acidic sources (H_2_SO_4_, H_3_PO_4_, Mg(HSO_4_)_2_) the use of BINOL-derived phosphoric acid as a catalyst is very convenient since it facilitates the purification process and has no interaction with the drying agent. The mechanism of the reaction comprises an initial formation of imine and enamine species **13** and **14** through a double condensation of the amine **1** with aldehyde **2** and pyruvate derivative **3**, followed by an acid promoted Mannich reaction that leads to the formation of intermediate **15**. Finally, an intramolecular cyclization between the amine and the ester groups yields γ-lactam substrates **4**–**12**.

Keeping into account the mechanism proposed for this transformation, the electronic character of the amine substrate might be the crucial factor in the reactivity of the reaction. While the use of activated amines may benefit the nucleophilic character of enamine nucleophiles **14** this would result in a decreased electrophilic character of imine substrates **13**. On the contrary, the use of deactivated amines would result in activation of imine electrophile **13** and deactivation of enamine nucleophile **14**.

Following this approach, 38 densely functionalized substrates were synthesized, to illustrate the synthetic potential of the reaction. First γ-lactams **4–5** were prepared using weakly activated *p*-toluidine (**1a**, R^1^ = *p*-CH_3_C_6_H_4_) as amine substrate, as well as diverse aldehydes and pyruvate derivatives ([Fig pharmaceuticals-14-00782-ch001]). Using ethyl pyruvate (**3a**, R^3^ = H), simple formaldehyde (**2a**, R^2^ = H) or benzaldehyde (**2b**, R^2^ = Ph), aromatic aldehydes bearing electron-donating and electron-withdrawing groups **2c–i**, heteroaryl and naphthyl carboxaldehydes **2j–m**, and simple aliphatic aldehydes **2n–q** were used successfully in the multicomponent reaction ([Fig pharmaceuticals-14-00782-ch001], **4a–q**). Moreover, other aldehydes such as cinnamaldehyde (**2r**, R^2^ = CH=CHPh) or ethyl glyoxalate (**2s**, R^2^ = CO_2_Et) were also effectively used as substrates ([Fig pharmaceuticals-14-00782-ch001], **4r–s**). Remarkably, fluorine-containing γ-lactam substrates were also synthesized using perfluorobenzaldehyde (**2t**, R^2^ = C_6_F_5_) or trifluoroacetaldehyde (**2u**, R^2^ = CF_3_) ([Fig pharmaceuticals-14-00782-ch001], **4t**,**u**), the latter generated in situ from its hydrate, and even phosphorus-containing substrates were efficiently obtained when β-phosphorated aldehydes **2v**,**w** (R^2^ = CH_2_P(O)(OEt)_2_, CH_2_P(O)Ph_2_) were used as starting materials in the multicomponent process ([Fig pharmaceuticals-14-00782-ch001], **4v**,**w**).

Additionally substituted ethyl pyruvate derivatives **3b–d** (R^3^ = Me, Bn, CO_2_Me) and weakly activated *p*-toluidine (**1a**, R^1^ = *p*-CH_3_C_6_H_4_) react with *p*-nitrobenzaldehyde (**2g**, R^2^ = *p*-NO_2_C_6_H_4_) or benzaldehyde (**2b**, R^2^ = Ph) to afford 4-substituted γ-lactam derivatives ([Fig pharmaceuticals-14-00782-ch001], **5a–c**). In this case, the reaction should be heated and performed in methyl *tert*-butyl ether as a solvent, which may be due to the high steric hindrance present in the final product. Under the same conditions β-phosphorated pyruvate derivatives **3e**,**f** (R^3^ = P(O)(OEt)_2_, P(O)OPh_2_) react with *p*-toluidine (**1a**, R^1^ = *p*-CH_3_C_6_H_4_) and formaldehyde (**2a**, R^2^ = H) yielding phosphorus substituted γ-lactams ([Fig pharmaceuticals-14-00782-ch001], **5d**,**e**).

Next, the reaction was extended to the use of other amine substrates. The reaction of ethyl pyruvate (**3a**, R^3^ = H) with electron-rich *p*-anisidine (**1b**, R^1^ = *p*-CH_3_OC_6_H_4_) and several aldehydes **2** affords the corresponding γ-lactam derivatives in very good yields ([Fig pharmaceuticals-14-00782-ch002], **6a–c**) while, if halogen-substituted anilines **1c–g** are used as amine substrates in the same reaction, γ-lactam derivatives are again efficiently obtained ([Fig pharmaceuticals-14-00782-ch002], **7–10**). The reaction even tolerates the use of deactivated aromatic amines such as *m*-trifluoromethylaniline (**1h**, R^1^ = *m*-CF_3_C_6_H_4_) for the formation of substrates containing trifluoromethyl groups ([Fig pharmaceuticals-14-00782-ch002], **11a**,**b**) and one example of the utilization of a heteroaromatic amine **1h** (R^1^ = 2-quinolinyl), that leads to the formation of a quinoline derived substrate, is also shown ([Fig pharmaceuticals-14-00782-ch002], **12**. In the latter case, a low yield of lactam substrate **12a** is obtained, possibly due to the high steric crowding present in the final structure.

Besides, when we tried to extend the multicomponent protocol to the use of ketone electrophiles instead of aldehydes, our reaction failed. In the case of the reaction using ethyl pyruvate (**3a**), and *p*-toluidine (**1a**), the imine species generated from pyruvate and the amine was found to be more reactive than the alternative ketimine species and a self-condensation reaction was observed. Accordingly, substrate **16** is obtained when the reaction is performed using one equivalent of *p*-toluidine (**1a**) and ethyl pyruvate (**3a**), where two units of each substrate conform to the structure of the γ-lactam derivative. Additionally, the hydrogenation of 1,5-dihydro-2*H*-pyrrol-2-one ring in **16** allowed the obtaining of saturated γ-lactam **17** in very good yield and as a single diastereoisomer (Scheme 2).

The relative configuration of the stereocenters in saturated γ-lactam **17** was determined by NOESY. NMR experiments with compound **17** showed NOE effect between both, the proton and the methyl group at the two stereogenic centers, at δ = 4.26 and 1.63 ppm, respectively, with the same NMR signal corresponding to one of the protons of the diastereotopic CH_2_ group at δ = 2.61 ppm, indicating that these three atoms are oriented in the same direction. As expected, a strong NOE is observed between both diastereotopic protons. The fact that almost no NOE is observed between the signal corresponding to the second of the diastereotopic protons, at δ = 2.46 ppm, and the proton and the methyl group at the stereogenic centers confirms a relative *cis* configuration between the carboxylate and the amino groups.

Then, to extend the diversity of our substrates, some synthetic transformations of 1,5-dihydro-2*H*-pyrrol-2-one substrates were performed. The hydrogenation of phosphorus and fluorine substituted γ-lactams **4u**,**v** and **6a**, in the presence of a palladium catalyst, affords saturated 5-membered ring substrates in excellent yields ([Fig pharmaceuticals-14-00782-ch003], **18a–c**). Likewise, treatment of γ-lactam **4v** with methyllithium followed by the addition of methyl iodide is known to give the dialkylated substrate ([Fig pharmaceuticals-14-00782-ch003], **19**), while the metalation of **4v** with lithium di-*iso*-propylamide (LDA) followed by the addition of an aldehyde affords conjugated azatrienes **20** through a Horner–Wadsworth–Emmons olefination reaction and a subsequent ring-opening of the γ-lactam core ([Fig pharmaceuticals-14-00782-ch003], **20a**,**b**).

All our other additional attempts to functionalize the enamine moiety of lactam substrate **4b** were unsuccessful but, surprisingly, when a bulky base like LDA was used, to generate the aza-enolate species, oxidation of the substrate was observed, leading to the formation of γ-lactam **21** that could be only isolated in very low yields (Scheme 3).

With this batch of structures in our hands, we faced the study of their biological activity. Accordingly, the antiproliferative activity of our γ-lactam derivatives against several cancer cell lines was next investigated.

### 2.2. Biological Results

In vitro cytotoxicity of the γ-lactam derivatives was evaluated by testing their antiproliferative activities against several human cancer cell lines. Cell counting kit (CCK-8) assay was used for the evaluation of growth inhibition. Moreover, non-malignant MRC5 lung fibroblasts were tested for studying selective toxicity [57] and chemotherapeutic doxorubicin is used as the reference value.

In a preliminary study, we tested the cytotoxicity of γ-lactam **4b** as lead compound against seven human cancer cell lines: HEK293 (human embryonic kidney), MCF7 (human breast adenocarcinoma), HTB81 (human prostate carcinoma), HeLa (human epithelioid cervix carcinoma), RKO (human colon epithelial carcinoma), SKOV3 (human ovarian carcinoma) and A549 (carcinomic human alveolar basal epithelial cell). The cell proliferation inhibitory activity of γ-lactam **4b** is shown as IC_50_ values (Figure 2). Although no grown inhibition activity was observed against HEK293 and MCF7 cell lines, moderate cytotoxicity was observed in the case of HTB81, HeLa, and RKO cell lines. Remarkably, promising IC_50_ values of γ-lactam **4b** against SKOV3 and A549 cell lines and, interestingly very good selectivity was also obtained towards MRC5 non-malignant cell line (Figure 2).

With these results in hand, we extended our study of the cell proliferation inhibitory activities of our family of γ-lactams using RKO, SKOV3, and A549 cell lines. The IC_50_ values of γ-lactam derivatives **4–12** ([Fig pharmaceuticals-14-00782-ch001] and [Fig pharmaceuticals-14-00782-ch002]) are presented in Table 1. First, γ-lactam derivatives **4** derived from *p*-toluidine were evaluated. The simplest member of this family is substrate **4a**, which is obtained from formaldehyde and holds no substituent at position 5 of the ring. Compound **4a** did not show activity against the SKOV3 cell line in vitro, but a slight activity was observed for the A549 cell line, with modest IC_50_ values of 38.25 ± 3.35 µM (Table 1, Entry 1). The cytotoxicity of γ-lactams **4** was found to be strongly dependent on the substituent at the 5 position of the ring. Phenyl substituted lactam **4b** showed improved IC_50_ values of = 9.62 ± 1.18 and 2.34 ± 0.28 µM against SKOV3 and A549, respectively, and good selectivity with respect to non-malignant lung fibroblasts (Table 1, Entry 2).

The introduction of methyl groups into a bioactive structure makes it more lipophilic, often resulting in an improved ability of molecules to cross cell membranes [58,59]. However, in this case, *ortho*, *meta* or *para* methyl-substituted benzene substituents in the position 5 of the ring resulted in a drop in the cytotoxic activity against the A549 cell line, with IC_50_ values around 20–40 µM, and a complete loss of the activity in SKOV3 cell line with IC_50_ values over 50 µM (Table 1, Entries-3–5).

Although, the effect of the introduction of fluorine atoms in the structure of organic compounds is rather difficult to predict, very often it leads to increased activities [60,61,62]. For this reason, next, we tested the in vitro cytotoxicity of fluorine-containing γ-lactams **4f**,**i** against SKOV3 and A549 cell lines. The introduction of a *para*-fluorophenyl substituent into the 5-membered ring did not result in improved activity and IC_50_ values of 23.52 ± 0.75 and 10.72 ± 1.21 µM were obtained in SKOV3 and A549 cell lines respectively for compound **4f** (Table 1, Entry 6). Similarly, *para*-trifluoromethylphenyl substituted γ-lactam **4i** showed IC_50_ values of over 50 µM for both cancer cell lines (Table 1, Entry 9). γ-Lactam substrates bearing other electron-poor aromatics such as nitrophenyl groups showed similar results with IC_50_ values of 27.65 ± 1.32 and 7.33 ± 0.57 µM in SKOV3 and A549 cell lines, respectively, for *para*-nitrophenyl substituted compound **4g** (Table 1, Entry 7) and values of 27.65 ± 1.32 and 7.33 ± 0.57 µM, for both cell lines, in the case of *meta*-nitrophenyl substituted γ-lactam **4h** (Table 1, Entry 8). Remarkably, compound **4g** did show some cytotoxicity against the RKO cell line with an IC_50_ value of 23.84 ± 2.53 µM but not much selectivity against healthy cells was observed in this case (Table 1, Entry 7). Moreover, switching the aromatic substituent by a heteroaromatic or naphthyl one did not have a positive effect on the cytotoxicity against SKOV3 and A549 cell lines since *N*-CH_3_-indolyl, furyl, thiophene, or naphthyl substituted lactam derivatives **4j-m** presented higher IC_50_ values if compared with the benzene substituted substrate **4b** (Table 1, Entries 10–13).

Next, we studied the introduction of aliphatic substituents at position 5 of the γ-lactam ring. The cytotoxic effect found for methyl-substituted lactam **4n** was found to be slightly lower if compared with phenyl substituted lactam **4b**, with IC_50_ values of 29.16 ± 1.00 and 15.68 ± 0.92 µM in SKOV3 and A549 cell lines, respectively (Table 1, Entry 14). Substitution with larger aliphatic substituents, such as *iso*-propyl, *iso*-butyl, or cyclohexyl groups, resulted in a complete loss of the cytotoxic activity of substrates **4o–q** against SKOV3 cell line and nearly the same IC_50_ values or slightly better against A549 cell line (Table 1, Entries 15–17). Substrate **4q** presented an IC_50_ value of 23.79 ± 1.32 µM against the RKO cell line. Besides, cinnamyl substituted γ-lactam **4r** showed modest IC_50_ values of about 10 µM for SKOV3 and A549 cell lines, but not much selectivity with respect to non-malignant cells, and ethoxycarbonyl substituted lactam **4s** did not show any cytotoxic activity (Table 1, Entries 18, 19).

Continuing with our interest in fluorine-containing molecules, next we faced the study of the effect of perfluoroalkylalted substituents on the antiproliferative activity of our γ-lactam substrates. Nevertheless, perfluorophenyl and trifluoromethyl substituted γ-lactams **4t**,**u** showed lower cytotoxic activity than their parent non-perfluorinated derivatives **4b** and **4n** (Table 1, Entries 2, 14 vs. 20, 21). In addition, it is well known that structural modifications of active molecules involving the introduction of phosphorus-containing functional groups, very often result in increased or new activities [63,64,65]. For this reason, next, we studied the cell proliferation inhibitory activities of phosphonate and phosphine oxide-derived substrates **4v**,**w**. Phosphonate derivative **4v** showed a modest cytotoxic activity, but only against A549 cell line, with an IC_50_ value of 11.29 ± 1.80 µM (Table 1, Entry 22) and, although phosphine oxide derivative **4w** presented cytotoxic properties against the three studied cancer cell lines, the selectivity with respect to non-malignant lung fibroblasts was not very significant (Table 1, Entry 23). 

After studying the effect of the substitution at position 5 of the ring into the antiproliferative activity of 1,5-dihydro-2*H*-pyrrol-2-ones **4**, we set up our following objective into the study of the effect of the substitution at the 4 position of 1,5-dihydro-2*H*-pyrrol-2-one ring. Accordingly, γ-lactams **5**, synthesized using substituted pyruvate derivatives, were next studied. Although none of those substrates showed any activity against the SKOV3 cell line, some interesting results were obtained with the A549 cell line. Methyl and benzyl substituted γ-lactams **5a**,**b** showed improved IC_50_ values of 2.05 ± 0.23 and 9.92 ± 1.15 µM (Table 1, Entries 24, 25) if compared with the parent unsubstituted derivative **4g** (Table 1, Entry 7). Remarkably, lactam **5c** holding a methyl carboxylate substituent at position 5 of the ring showed a very good IC_50_ value of 1.67 ± 0.49 µM against A549 cell line and a very good selectivity if compared with SKOV3 or healthy cells. Moreover, the introduction of a phosphonate substituent at the 4 position results in a dramatic decrease in the cytotoxic power in γ-lactam **5d** with an IC_50_ value higher than 50 µM (Table 1, Entry 27), which is reflected by the comparison with the IC_50_ value observed in its unsubstituted derivative **4a** (Table 1, Entry 1). Unfortunately, the activity of diphenylphosphine oxide-derived compound **5e** could not be tested due to the low solubility of the substrate in the solvents that are compatible with the tests for in vitro assays (Table 1, Entry 28).

Once the influence of the substituents at the positions 1, 3, and 4 of the γ-lactam ring into the cell proliferation inhibitory activities of our substrates have been described, the next point to be addressed is the impact of the amine substituents in the growth inhibition activity of lactams **6–12** ([Fig pharmaceuticals-14-00782-ch002]). First, *p*-anisidine derived substrates **6** were examined. Phenyl substituted lactam **6a** showed improved cytotoxicity against the SKOV3 cell line but slightly worse against the A549 cell line than its parent *p*-toluidine derivative **4b** with IC_50_ values of 6.84 ± 0.59 and 12.02 ± 1.96 µM. Their selectivity with respect to non-malignant cells was very satisfactory (Table 2, Entry 1 vs. Table 1, Entry 2). *p*-Nitrophenyl substituted derivative **6b** did not show any improvement with respect to derivative **4h** (Table 2, Entry 2 vs. Table 1 entry 7) and phosphonate derivative **6c** did not show any cytotoxicity against SKOV cell line but showed improved IC_50_ values than its parent substrate **4v** of 6.84 ± 0.22 µM against A549 cells (Table 2, Entry 3 vs. Table 1 entry 22).

Next, we studied the effect of the introduction of halogen atoms in the structure of the amine moiety of substrates **7–10**. No significant changes were observed in the cytotoxic effect of *p*-chloroaniline derivative **7**, holding a *p*-nitrophenyl at the 5 position of the ring, if compared to its *p*-toluidine or *p*-anisidine derivative **4g** or **6b** (Table 2, Entry 4 vs. Table 1, Entry 7 and Table 2, Entry 2). Likewise, *p*-bromoaniline derivatives **8a**,**b** did not show any grown inhibition in any of the cell lines (Table 2, Entries 5, 6). Switching the chlorine atom from *para* to the *meta* position in the aromatic ring of the amine improved the cytotoxic activity in both cell lines for γ-lactam **9**, although in this case, the selectivity against non-malignant cells gets worse (Table 2, Entry 7). Very good results, however, were obtained for *o*-fluoroaniline derived γ-lactam **10** in terms of cytotoxicity and selectivity, with IC_50_ values of 5.55 ± 0.62 and 4.36 ± 0.51 µM in SKOV3 and A549 cell lines, respectively, substantially improving the values observed for its non-fluorinated derivative **4g** (Table 2, Entry 8 vs. Table 1, Entry 7). The use of *m*-trifluoromethyl substituted aniline substrates resulted also in an improvement of the grown inhibition of compounds **11a**,**b**. Unsubstituted formaldehyde derivative **11a** showed slightly better IC_50_ values than the model non-fluorinated derivative **4a** (Table 2, Entry 9 vs. Table 1, Entry 1) but substrate **11b** holding a *p*-nitrophenyl substituent at the position 5 of the lactam ring showed very good IC_50_ values of 8.50 ± 0.54 and 2.00 ± 0.78 µM in SKOV3 and A549 cell lines, respectively, significantly lower than the values observed for the parent *p*-toluidine derived substrate **4g** (Table 2, Entry 10 vs. Table 1, Entry 7).

The effect of a heteroaromatic amine moiety on the γ-lactam structure was also studied. In this context, quinoline scaffold plays an important role in drug development [66,67] and many of their derivatives have shown their potential as anticancer agents [68] such as well-known camptothecin and their derivatives [69]. However, although an improvement in the cytotoxic activity and selectivity in the A549 cell line was observed in 2-quinolinylamine substituted lactam **12a** with respect to other similar derivatives, like *p*-toluidine, *p*-anisidine, *p*-chloroaniline, and *p*-bromoaniline substituted lactams **4g**, **6b**, **7**, and **8a** (Table 2, Entry 11 vs. Table 1, Entries 7 and Table 2, Entries 2, 4 and 5), *m*-chloroaniline, *o*-fluoroaniline, and *m*-trifluoromethylaniline substituted substrates **9**, **10** and **11b** proved to be superior to 2-quinolinylamine derivative **12a** in this respect (Table 2, Entries 7, 8 and 10 vs. Table 2, Entry 11). No activity against the SKOV3 cell line was observed however for substrate **12a**. 

To complete our research, we finally tested the cell proliferation inhibitory activities of other γ-lactam derivatives such as dimeric substrate **16** (see Scheme 2, vide supra) as well as derivatives **17–21**, obtained by the modification of some of the γ-lactam derivatives ([Fig pharmaceuticals-14-00782-ch003] and Scheme 3, vide supra).

Dimeric lactam derivative **16** showed a modest IC_50_ value 13.97 ± 1.05 µM in the A549 cell line, and it presented a high selectivity if compared with SKOV3, RKO, or MRC5 cell lines (Table 3, Entry 1). Unfortunately, the reduction of compound **16** showed to have a negative effect on the cytotoxicity of lactam derivatives, and no activity was observed for the saturated γ-lactam derivative **17** (Table 3, Entry 2). With respect to the activity against A549 cell line, the reduction of dihydropyrrolone ring in compounds **4u,v** resulted in an improvement in the cytotoxic activity of *p*-toluidine derived γ-lactam compounds **18a,b** with better IC_50_ values of 3.27 ± 0.65 and 4.25 ± 0.65 µM, respectively (Table 3, Entries 3, 4 vs. Table 1, Entries 21, 22). However, in the case of *p*-anisidine derivative **18c**, a slightly worse IC_50_ value of 19.45 ± 0.34 µM was found if compared with its parent derivative **6a** (Table 3, Entry 5 vs. Table 2, Entry 1). With respect to the SKOV3 cell line, the reduction of dihydropyrrolone ring in compounds **4u,v** did not result in any improvement into the grown inhibition activity, and γ-lactam derivatives **18a,b** presented IC_50_ values higher than 50 µM, as their parent unsaturated compounds **4u,v** (Table 3, Entries 3, 4 vs. Table 1, Entries 21, 22). Although saturated γ-lactam derived from *p*-anisidine **18c** still showed an IC_50_ value of 19.45 ± 0.34 µM, this value was found to be slightly higher than its unsaturated parent compound **6a** (Table 3, Entry 5 vs. Table 2, Entry 1).

As it has been addressed above, due to the lipophilic character of alkyl groups, the functionalization of bioactive structures with methyl groups often results in stronger activities than the parent structures [59]. For that reason, we had high expectations with respect to the activity of lactam derivative **19** that holds four methyl groups unto its structure. Indeed, a slightly higher IC_50_ value of 3.65 ± 0.25 µM against A549 cell line was obtained for functionalized compound **19** with respect to the parent lactam **4v** (Table 3, Entry 6 vs. Table 1, Entry 22) but, more important, a highly selective behavior was observed for this compound that did not show toxicity against SKOV3, RKO and MRC5 cell lines (Table 3, Entry 6).

The conjugated compounds **20a**,**b**, obtained by the ring-opening of lactam **4v** presented good cytotoxicity against the A549 cell line, showing IC_50_ values of 6.90 ± 0.73 and 9.77 ± 0.17 µM, respectively, and a very good selectivity if compared with SKOV3 and healthy cells (MRC5) (Table 3, Entries 7, 8). In addition, oxidized lactam derivative **21** showed a very high toxicity against the A549 cell line and moderate activity against the SKOV3 cell line with IC_50_ values of 0.60 ± 0.08 and 16.72 ± 1.1 µM, respectively. However, compound **21** exhibited also a strong toxicity against healthy cells (Table 3, Entry 9). 

Finally, due to the presence of a stereogenic carbon in the structure of our lactam derivatives, we were intrigued about the real activity of the individual enantiomers. For that reason, we performed the enantioselective synthesis of lactam **4v**, using both enantiopure isomers of a substituted BINOL phosphoric acid catalyst in the multicomponent reaction (Scheme 4).

Both enantiomers of **4v** were prepared in 90% ee and, then, two almost enantiopure samples were obtained in 97% ee by a subsequent crystallization. However, no significant differences were observed for *R* or *S* enantiomers if compared with the racemic sample with IC_50_ values of 9.23 ± 1.68 and 10.53 ± 0.80 µM, respectively.

Next, cell morphology analysis and flow cytometry assays were performed, to elucidate the mechanism of the antiproliferative activity of γ-lactam substrates. Cell morphology was analyzed at different time points after the addition of compound **4b** to visualize the cellular modifications as a result of the treatment. The first concentration studied was 1 µM, which is below the IC_50_ of **4b** and, accordingly, cells showed a healthy, uniform morphology and cellular growth was recognizable over time, meaning that this concentration was well tolerated by the cells (Figure 3A). On the contrary, after the addition of 5 µM of **4b**, alterations of the cellular morphology were observed, as well as a mild disruption of the cellular growth, especially after 6 h of exposure (Figure 3B). At higher concentrations, the alterations observed in cell morphology clearly evidenced high cytotoxicity (Figure 3C,D). These results support previously obtained data in relation to the cytotoxicity and the IC_50_ of **4b**.

In addition, flow cytometry assays were performed using compound **4b** in A549 cells at 12, 24, and 48 h post-exposure. Early apoptotic cells were FL-1 positive and FL-3 negative, meaning that they had initialized the apoptotic process but were still alive at the time of the measurement. Percentages of these early-apoptotic cells after 12, 24, and 48 h. were, respectively, 31.17 ± 2.04%, 34.97 ± 3.62% and 40.13 ± 0.81%. On the other hand, late apoptotic and necrotic cells were both FL-1 and FL-3 positive, meaning that they activated the apoptotic machinery and were dead at the time of the measurement. Percentages of late-apoptotic and necrotic cells after 12, 24 and 48 h were, respectively, 33.03 ± 1.43%, 37.60 ± 3.37% and 36.90 ± 0.70% (Figure 4). Control live cells were all FL-1 and FL-3 negative, while around 95% of control necrotic cells were FL-3 positive, FL-1 negative (See ESI). Taken together, these results suggest that one of the main mechanisms by which **4b** induces cytotoxicity is based on the activation of intracellular apoptotic mechanisms. In fact, among all the dead cells detected, which were FL-1 positive, almost all were simultaneously positive for the FL-3 channel too, while only a small percentage (2% or lower) of dead cells were exclusively FL-1 positive. Therefore, the majority of dead cells detected in this study were apoptotic cells. In addition, the total percentages of FL-3 positive cells, which comprised early and late apoptotic cells, were over 60%, and the percentage increased over time.

It is also worth mentioning that, except for substrates **8a**, **8b**, and **11b**, all other γ-lactam derivatives fulfill the requirements for orally active drugs in humans, in accordance with Lipinski’s rule of five. According to the predictions, most of the described substrates have a high gastrointestinal absorption and the ability to cross the blood–brain barrier (See Appendix A).

## 3. Material and Methods

### 3.1. Chemistry

#### 3.1.1. General Experimental Information

Solvents for extraction and chromatography were technical grade. All solvents used in reactions were freshly distilled from appropriate drying agents before use. All other reagents were recrystallized or distilled as necessary. All reactions were performed under an atmosphere of dry nitrogen. Analytical TLC was performed with silica gel 60 F_254_ plates. Visualization was accomplished by UV light. ^1^H, ^13^C, ^31^P, and ^19^F NMR spectra were recorded on a Varian Unity Plus (Varian Inc, NMR Systems, Palo Alto (CA), United States) (at 300 MHz, 75 MHz, 120 MHz, and 282 MHz respectively) and on a Bruker Avance 400 (Bruker BioSpin GmbH, Rheinstetten, Germany) (at 400 MHz for ^1^H, and 100 MHz for ^13^C). Chemical shifts (δ) are reported in ppm relative to residual CHCl_3_ (δ = 7.26 ppm for ^1^H and δ = 77.16 ppm for ^13^C NMR) and using phosphoric acid (50%) as external reference (δ = 0.0 ppm) for ^31^P NMR spectra. Coupling constants *(J)* are reported in Hertz. Data for ^1^H NMR spectra are reported as follows: chemical shift, multiplicity, coupling constant, integration. Multiplicity abbreviations are as follows: s = singlet, d = doublet, t = triplet, q = quartet, m = multiplet). ^13^C NMR peak assignments were supported by Distortionless Enhanced Polarization Transfer (DEPT). High-resolution mass spectra (HRMS) were obtained by positive-ion electrospray ionization (ESI). Data are reported in the form *m/z* (intensity relative to base = 100). Infrared spectra (IR) were taken in a Nicolet iS10 Thermo Scientific spectrometer (Thermo Scientific Inc., Waltham, Massachusetts (MA) as neat solids. Peaks are reported in cm^–1^. Phosphorated aldehydes (diethyl (2-oxoethyl)phosphonate and 2-(diphenylphosphoryl)acetaldehyde), phosphorated pyruvates (ethyl 3-(diethoxyphosphoryl)-2-oxopropanoate and ethyl 3-(diphenylphosphoryl)-2-oxopropanoate) and 1-ethyl 4-methyl 2-oxosuccinate were synthesized following literature procedures [70,71,72,73].

#### 3.1.2. Compounds Purity Analysis

All synthesized compounds were analyzed by HPLC to determine their purity. The analyses were performed on Agilent 1260 Infinity HPLC system (Agilent, Santa Clara, CA, United States) (C-18 column, Hypersil, BDS, 5 μm, 0.4 mm × 25 mm) at room temperature. All the tested compounds were dissolved in dichloromethane, and 5 μL of the sample was loaded onto the column. Ethanol and heptane were used as the mobile phase, and the flow rate was set at 1.0 mL/min. The maximal absorbance at the range of 190−400 nm was used as the detection wavelength. The purity of all the tested lactam derivatives **4**–**12** and **16**–**21** is >95%, which meets the purity requirement by the Journal.

#### 3.1.3. Representative Experimental Procedures and Characterization Data for Compounds **4**–**12** and **16**–**21**

##### Representative Procedure for the Multicomponent Reaction of Amines **1**, Aldehydes **2**, and Pyruvate Derivatives **3**

A solution of amine **1** (4 mmol), aldehyde **2** (2 mmol), pyruvate derivative **3** (6 mmol), and BINOL derived phosphoric acid (70 mg, 0.2 mmol) was stirred in diethyl ether or MTBE (10 mL) in the presence of anhydrous MgSO_4_ at room temperature or 55 °C for 48 h (see ESI). The volatiles were distilled off at reduced pressure and the crude residue was purified by crystallization in diethyl ether or by column chromatography (Hexanes/AcOEt) to afford pure lactams **4–12**.

*1-(p-Tolyl)-3-(p-tolylamino)-1,5-dihydro-2H-pyrrol-2-one* (**4a**). The general procedure was followed, using *p*-toluidine (429 mg, 4 mmol), 37% aq. formaldehyde (150 μL, 2 mmol) and ethyl pyruvate (670 μL, 6 mmol) in diethyl ether at room temperature and the residue was purified by column chromatography (Hexanes/AcOEt 9:1) affording 445 mg (80%) of **4a** as an orange solid. M.p. (Et_2_O) = 178–180 °C. ^1^H NMR (400 MHz, CDCl_3_): δ 7.64 (d, ^3^*J*_HH_ = 8.5 Hz, 2H), 7.20 (d, ^3^*J*_HH_ = 8.2 Hz, 2H), 7.13 (d, ^3^*J*_HH_ = 8.2 Hz, 2H), 7.1 (d, ^3^*J*_HH_ = 8.5 Hz, 2H), 6.53 (s, 1H), 5.97 (t, ^3^*J*_HH_ = 2.6 Hz, 1H), 4.37 (d, ^3^*J*_HH_ = 2.6 Hz, 2H), 2.34 (s, 3H), 2.32 (s, 3H). ^13^C {^1^H} NMR (101 MHz, CDCl_3_): δ 166.5 (C=O), 139.2 (C_quat_), 136.8 (C_quat_), 134.5 (C_quat_), 134.3 (C_quat_), 130.7 (=C_quat_), 130.0 (2xCH), 129.8 (2xCH), 119.0 (2xCH), 116.8 (2xCH), 99.8 (CH), 49.8 (CH_2_), 21.0 (CH_3_), 20.8 (CH_3_). FTIR (neat) ν_max_: 3325 (N-H), 1671 (C=O), 1644 (C=CH). HRMS (ESI-TOF) m/z calcd for C_18_H_18_N_2_O [M + H]^+^ 279.1497, found 279,1501.

##### Synthesis of Ethyl 2-methyl-5-oxo-1-(p-tolyl)-4-(p-tolylamino)-2,5-dihydro-1*H*-pyrrole-2-carboxylate (16)

A solution of ethyl pyruvate (445 mg, 4 mmol), *p*-toluidine (429 mg, 4 mmol), and BINOL derived phosphoric acid catalyst (0.2 mmol) was stirred in DCM (10 mL) in the presence of anhydrous MgSO_4_ at room temperature for 48h. The volatiles were distilled off at reduced pressure and the crude residue was purified by column chromatography (Hexanes/AcOEt 8:2) to afford 496 mg (79%) of **16** as a white solid. M.p. (Et_2_O) = 188–190 °C (dec.). ^1^H NMR (300 MHz, CDCl_3_) δ 7.26–7.19 (m, 4H, 4H, 4xCH_Ar_), 7.15 (d, ^3^*J_HH_* = 8.2 Hz, 2H, 2xCH_Ar_), 6.99 (d, ^3^*J_HH_* = 8.3 Hz, 2H, 2xCH_Ar_), 6.59 (s, 1H, NH), 5.92 (s, 1H, CHN), 4.31–4.06 (m, 2H, CH_2_CH_3_), 2.37 (s, 3H, CH_3_ Tol), 2.33 (s, 3H, CH_3_ Tol), 1.64 (s, 3H, CH_3_), 1.25 (t, ^3^*J_HH_* = 7.1 Hz, 3H, CH_2_CH_3_).^13^C {^1^H} NMR (75 MHz, CDCl_3_): δ 171.9 (C=O), 167.8 (C=O), 138.8 (C_quat_), 137.0, (C_quat_), 134.2, (C_quat_), 133.5, (C_quat_), 131.3, (C_quat_), 130.1, (2xCH_Ar_), 130.0 (2xCH_Ar_), 126.0 (2xCH_Ar_), 117.3 (2xCH_Ar_), 107.0 (=CH-), 68.9 (C_quat_-N), 62.3 (CH_2_), 21.5 (CH_3_), 21.3 (CH_3_), 20.9 (CH_3_), 14.3 (CH_3_). FTIR (neat) ν_max_: 3305 (N-H), 1705 (C=O), 1679 (C=O), 1640 (-C=CH). HRMS (ESI-TOF) m/z calcd for C_22_H_25_N_2_O_3_ [M + H]^+^ 365,1865, found 365,1860.

##### Representative Procedure for the Hydrogenation of γ-Lactams **4**, **6**, and **16**

A mixture of γ-lactam (0.5 mmol) and 52 mg of 10% palladium on carbon (0.05 mmol Pd) in methanol (100 mL) was stirred for 10 h under hydrogen pressure at 80 psi. The reaction mixture was filtered through celite, and the organic fractions were distilled off at reduced pressure. The resulting residue was crystallized in methanol to afford pure lactams **17** and **18a–c.**

*Ethyl (2S*, 4R*)-2-methyl-5-oxo-1-(p-tolyl)-4-(p-tolylamino)-pyrrolidine-2-carboxylate* (**17**). The general procedure was followed, using ethyl 2-methyl-5-oxo-1-(*p*-tolyl)-4-(*p*-tolylamino)-2,5-dihydro-1*H*-pyrrole-2-carboxylate (**16**) (182 mg, 0.5 mmol) affording 159 mg (87%) of **17** as a white solid. M.p. (Et_2_O) = 173–174 °C. ^1^H NMR (300 MHz, CDCl_3_) δ 7.21 (d, ^3^*J_HH_* = 8.4 Hz, 2H), 7.15 (d, ^3^*J_HH_* = 8.4 Hz, 2H), 7.02 (d, ^3^*J_HH_* = 8.1 Hz, 2H), 6.60 (d, ^3^*J_HH_* = 8.1 Hz, 2H), 4.42 (bs, 1H), 4.26 (m, 1H), 4.15 (q, ^3^*J_HH_* = 7.0, 2H), 2.61 (m, 1H), 2.46 (m, 1H), 2.36 (s, 3H), 2.26 (s, 3H), 1.63 (s, 3H), 1.21 (t, ^3^*J_HH_* = 7.0 Hz, 3H).^13^C {^1^H} NMR (75 MHz, CDCl_3_): δ 173.8 (C=O), 173.3 (C=O), 144.9 (C_quat_), 138.2 (C_quat_), 133.6 (C_quat_), 130.1 (2xCH), 130.0 (2xCH), 128.0 (C_quat_), 127.8 (2xCH), 114.1 (2xCH), 66.4 (C_quat_), 62.2 (CH_2_), 54.4 (CH), 41.7 (CH_2_), 23.5 (CH_3_), 21.3 (CH_3_), 20.6 (CH_3_), 14.2 (CH_3_). FTIR (neat) ν_max_: 3343 (N–H), 1701 (C=O), 1685 (C=O). HRMS (ESI-TOF) m/z calcd for C_22_H_27_N_2_O_3_ [M + H]^+^ 367,2021, found 367,2016.

##### Synthesis of Diethyl ((3,4-dimethyl-5-oxo-1-(p-tolyl)-4-(p-tolylamino)-4,5-dihydro-1*H*-pyrrol-2-yl)methyl)phosphonate (**19**)

A solution of lactam **4v** (214 mg, 0.5 mmol) in THF was cooled to −78 °C, methyl lithium (700 μL, 1.6M, 1.1 mmol) was dropwise added in the mixture and stirred for 1 h. After 1 h at −78 °C a solution of methyl iodide (70 μL, 1.1 mmol) was slowly added and the reaction was warmed to room temperature overnight. Water (5 mL) was added and the resulting mixture was extracted with AcOEt (3 × 10 mL). The combined organic phases were dried with anhydrous MgSO_4_ and concentrated under reduced pressure. The crude residue was purified by column chromatography (Hexanes/AcOEt 7:3) affording 196 mg (87%) of **19** as a yellow solid. Physical and spectroscopic data are in agreement with literature data [40].

##### Representative Procedure for the Horner-Wadsworth-Emmons Reactions of γ-Lactam 4v

A solution of lactam **4v** (214 mg, 0.5 mmol) in THF (2 mL) was dropwise added to a cooled solution (−78 °C) of LDA (1.1 mmol, prepared from 690 μL of 1.6 M *n*-butyl lithium and 150 μL of diisopropylamine in 5 mL of THF) and the mixture was stirred for 1 h at −78 °C. Then, the corresponding aldehyde (0.75 mmol) was added and the reaction was warmed to room temperature overnight. The solution was quenched with 10 mL of water and extracted with dichloromethane (3 × 10 mL), dried with anhydrous MgSO_4_, and concentrated at reduced pressure. The crude residue was purified by column chromatography (Hexanes/AcOEt 9:1) to afford products **20a–b**.

*(2Z, 3E, 5E)-6-Phenyl-N-(p-tolyl)-2-(p-tolyl imino)hexa-3,5-dienamide* (**20a**). The general procedure was followed, using benzaldehyde (76 μL, 0.75 mmol) affording 172 mg (90%) of **20a** as a yellow solid. Physical and spectroscopic data are in agreement with literature data [40].

##### Synthesis of (Z)-5-phenyl-1-(p-tolyl)-3-(p-tolylimino)-1,3-dihydro-2*H*-pyrrol-2-one (**21**)

A solution of lactam **4b** (214 mg, 0.5 mmol) in THF (2 mL) was dropwise added to a cooled solution (−78 °C) of LDA (1.1 mmol, prepared from 690 μL of 1.6 M *n*-butyl lithium and 150 μL of diisopropylamine in 5 mL of THF) and the mixture was stirred for 1 h. at −78 °C and the reaction was warmed to room temperature overnight. The solution was quenched with 10 mL of water and extracted with dichloromethane (3 × 10 mL), dried with anhydrous MgSO_4_ and concentrated at reduced pressure. The crude residue was purified by column chromatography (Hexanes/AcOEt 9:1) to afford 23 mg (13%) of **21** as an orange solid. M.p. (Et_2_O) = 207 °C (dec.).^1^H NMR (400 MHz, CDCl_3_) δ δ 7.35 (m, 1H, CH_Ar_), 7.27 (d, ^3^*J_HH_* = 7.8 Hz, 2H, 2xCH_Ar_), 7.23–7.20 (m, 4H, 4xCH_Ar_), 7.15–7.13 (m, 4H, 4xCH_Ar_), 6.99 (d, ^3^*J_HH_* = 8.3 Hz, 2H, 2xCH_Ar_), 6.01 (s, 1H, CH), 2.38 (s, 3H, CH_3_ Tol), 2.33 (s, 3H, CH_3_ Tol). ^13^C {^1^H} NMR (101 MHz, CDCl_3_): δ 165.3 (C=O), 159.5 (C_quat_), 156.3 (C_quat_), 147.9 (C_quat_), 137.5 (C_quat_), 136.5 (C_quat_), 132.2 (C_quat_), 130.7 (CH_Ar_), 129.9 (C_quat_), 129.8 (2xCH_Ar_), 129.7 (2xCH_Ar_), 128.6 (2xCH_Ar_), 128.2 (2xCH_Ar_), 127.0 (2xCH_Ar_), 122.3 (2xCH_Ar_), 97.1 (CH), 21.3 (2xCH_3_). FTIR (neat) ν_max_: 1690 (C=O), 1635 (C=N), 1640 (-C=CH). HRMS (ESI-TOF) *m*/*z* calcd for C_24_H_21_N_2_O [M + H]^+^ 353,1662, found 353.1740.

### 3.2. Biology

#### 3.2.1. Materials

Reagents and solvents were used as purchased without further purification. All stock solutions of the investigated compounds were prepared by dissolving the powdered materials in appropriate amounts of DMSO. The final concentration of DMSO never exceeded 5% (*v*/*v*) in reactions. The stock solution was stored at 5 °C until it was used.

#### 3.2.2. Cell Culture

Human epithelial lung carcinoma cells (A549) (ATCC^®^ CCL-185™, ATCC-Manassas, VA, USA) were grown in Kaighn’s Modification of Ham’s F-12 Medium (ATCC^®^ 30-2004™, ATCC-Manassas, VA, USA) and lung fibroblast cells (MRC5) (ATCC^®^ CCL-171™, ATCC-Manassas, VA, USA) were grown in Eagle’s Minimum Essential Medium (EMEM, ATCC^®^ 30-2003™, ATCC-Manassas, VA, USA). Epithelial ovary adenocarcinoma cells (SKOV3) (ATCC^®^ HTB-77™, ATCC-Manassas, VA, USA) were grown in McCoy’s 5A medium (ATCC^®^ 30-2007™, ATCC-Manassas, VA, USA) and colon carcinoma cells (RKO) (ATCC^®^ CRL-2577™, ATCC-Manassas, VA, USA) were grown in Eagle’s Minimum Essential Medium (EMEM, ATCC^®^ 30-2003™, ATCC-Manassas, VA, USA). All were supplemented with 10% of fetal bovine serum (FBS) (Sigma-Aldrich, Madrid, Spain) and with 1% of NORMOCIN solution (Thermo Fisher, Waltham, Massachusetts, MA, USA). Cells were incubated at 37 °C and 5% CO_2_ atmosphere and were split every 3–4 days to maintain monolayer coverage. For cytotoxicity experiments, A549 cells were seeded in 96-well plates at a density of 2.5–3 × 10^3^ cells per well and incubated overnight to achieve 70% of confluence at the time of exposition to the cytotoxic compound.

#### 3.2.3. Cytotoxicity Assays

Human epithelial lung carcinoma cells (A549) (ATCC^®^ CCL-185™, ATCC-Manassas, VA, USA) were grown in Kaighn’s Modification of Ham’s F-12 Medium (ATCC^®^ 30-2004™, ATCC-Manassas, VA, USA) and lung fibroblast cells (MRC5) (ATCC^®^ CCL-171™, ATCC-Manassas, VA, USA) were grown in Eagle’s Minimum Essential Medium (EMEM, ATCC^®^ 30-2003™, ATCC-Manassas, VA, USA). Both were supplemented with 10% of fetal bovine serum (FBS) (Sigma-Aldrich, Spain) and with 1% of NORMOCIN solution (Thermo Fisher, Waltham, Massachusetts, MA, USA). Cells were incubated at 37 °C and 5% CO_2_ atmosphere and were split every 3–4 days to maintain monolayer coverage. For cytotoxicity experiments, A549 cells were seeded in 96-well plates at a density of 2.5–3 × 10^3^ cells per well and incubated overnight to achieve 70% of confluence at the time of exposition to the cytotoxic compound.

#### 3.2.4. Evaluation of Cytotoxicity Mechanisms

Flow cytometry assays were conducted using a FACSCalibur system flow cytometer (Becton Dickinson Bioscience, Franklin Lakes, NJ, USA), to identify apoptotic cells and to differentiate them from necrotic cells. A549 cells were exposed to 5 µM of the cytotoxic compound **4b** and cell apoptosis and necrosis were evaluated 12, 24, and 48 h after exposure. For that purpose, treated cells were washed with phosphate-buffered saline (PBS) (Sigma-Aldrich, Spain) and detached with trypsin/EDTA (0.25%) (Gibco, New York, NY, USA). Cells were centrifuged at 1100 rpm for 5 min, and then the resulting pellet was resuspended in cell growth media and transferred to specific flow cytometer tubes. Propidium iodide (Sigma-Aldrich, Spain) at 1:300 dilution was used in each sample to detect necrotic cells and eBioscience™ Annexin V Apoptosis Detection Kit FITC (Fisher Scientific, Spain) was used to detect apoptotic cells following the manufacturer’s instructions. The fluorescent signals corresponding to necrotic cells and apoptotic cells were measured at 650 nm (FL3) and 525 nm (FL1), respectively. Non-treated cells, used as negative control samples, were displayed on a forward scatter (FSC) versus side scatter (SSC) dot plot to establish a collection gate and exclude cells debris. Cells treated with 1 µM of camptothecin (Sigma-Aldrich, Spain) served as a positive control for apoptosis and were used to establish cytometer settings and channel compensations. The experiments were carried out in triplicate for each condition. For each sample, 10,000 events were collected.

#### 3.2.5. Visualization of Cell Growth and Morphology

Qualitative analysis of A549 cell growth and morphology after exposure to 1 µM and 5 µM of the cytotoxic compound **4b** was conducted using Cytation^TM^ 1 Cell Imaging Multi-Mode Reader (Biotek, Winooski, VT, USA). Cell images were acquired immediately after the addition of the compound and at the following time-points after exposure: 0.5, 1, 1.5, 2, 3, 4, 8, 12, 16, 24, and 48 h.

## 4. Conclusions

In conclusion, we report an efficient multicomponent methodology for the preparation of 1,5-dihydro-2*H*-pyrrol-2-ones holding a variety of substituents at the five-membered ring. This strategy allows the possibility of assorted structural diversity in the resultant scaffold depending on the starting amine, aldehyde, and pyruvate derivative in a single step. Moreover, obtained lactam derivatives showed in vitro cytotoxicity inhibiting the growth of human tumor cell lines HTB81 (human prostate carcinoma), HeLa (human epithelioid cervix carcinoma), RKO (human colon epithelial carcinoma), SKOV3 (human ovarian carcinoma), and A549 (carcinomic human alveolar basal epithelial cell), and high selectivity toward MRC5 non-malignant lung fibroblasts. QSAR studies indicate that the cytotoxicity is enhanced, in general, by the presence of aromatic groups bearing lipophilic methyl substituents or fluorine atoms. Moreover, better cytotoxic activity is observed when the γ-lactam core is substituted by a polar ester group at position 4. Compound **5c** showed a very good IC_50_ value of 1.67 µM against A549 a cell line, and a very high selectivity if compared with the MRC5 cell line. Moreover, γ-lactam **10**, derived from *m*-fluoroalinine, presented IC_50_ values of 4.36 and 5.55 µM against A549 and SKOV3 cell lines and very good selectivity, with respect to healthy cells, and substrates **18a** and **19**, obtained by modification of the lactam structure, display very good IC_50_ values of 3.27 and 3.65 µM against A549. Although HTB81, HeLa, and RKO cell lines were proved to be more resistant to γ-lactam derivatives, compound **4b** presented IC_50_ values of 27.93, 30.23, and 22.07 µM for each cell line respectively and good selectivity towards MRC5 cell line. The main mechanism by which γ-lactams derivatives induce cytotoxicity is based on the activation of intracellular apoptotic mechanisms.

## Data Availability

The data presented in this study are available in the Appendix A or on request from the corresponding author (^1^H, ^13^C, ^19^F, and ^31^P-NMR and HRMS spectra and cytotoxicity essays).

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
