# Peer review of "A Multicomponent Protocol for the Synthesis of Highly Functionalized γ-Lactam Derivatives and Their Applications as Antiproliferative Agents"

_pharmaceuticals, 2021, doi:10.3390/ph14080782_

Round 1

Reviewer 1 Report

The manuscript titled “A multicomponent protocol for the synthesis of highly functionalized gamma-lactam derivatives and their applications as anti-proliferative agents” reports a multicomponent synthetic approach to quickly access different analogues of 3-amino 1,5-dihydro-2H-pyrrol-2-ones and their structure-activity relationship study as an anti-proliferative agent. The authors’ synthesized around 49 different analogues with varying substitution patterns at 1, 3, 4, and 5 positions of the gamma-lactam core and studied their antiproliferative activity against a host of cancer cell lines. Most of the analogues showed cytotoxicity against carcinomic human alveolar basal epithelial cells (A549). The aqueous solubility of some of the analogues is a concern that the authors’ can explore in their future work.  The authors’ used the flow cytometry technique to get some insight into the mechanism of cell death induced by the gamma-lactam analogues.  Full experimental and characterization details have been provided in the paper with appropriate HRMS and NMR analysis data. The work should be of interest to a wide group of medicinal chemists.

Considering the novelty of the analogues and their potential application as an antibacterial, I would recommend the manuscript for publication in the journal pharmaceuticals after following modifications.

Suggested corrections    

  1. Line 46: correct the word “through”.
  2. Scheme 1, 2, and 4: Reformat scrambled reagents and increase their font size.
  3. Indicate ring numbering for the lactam core of at least one analogue in chart 1 and chart 3.
  4. Increase the font size of the compound numbers in charts 1 and 2.
  5. Lactam ring numbering.
  6. Line 241: authors’ write; …remarkably, 4g…. However, in terms of cytotoxicity 4g seems to be similar to 4h.
  7. Line 273: …selectivity with respect to non…
  8. Compound 6c showed improved IC50 value against A549 cells.
  9. Line 372: …good selectivity compared to SKOV3 cell line and healthy cells (MRC5).

10. line 385: correct the word “significant”. 

Author Response

  1. Line 46: correct the word “through”.

Corrected

  1. Scheme 1, 2, and 4: Reformat scrambled reagents and increase their font size.

Schemes 1,2 and 4 have been reformatted. Structures in scheme 4 have been rearranged in order to fit in the page at the maximum size. Figure 1 and 2 and Scheme 3 have been also modified.

  1. Indicate ring numbering for the lactam core of at least one analogue in chart 1 and chart 3.

Ring numbering for the lactam core is indicated in Charts 1, 2 and 3.

  1. Increase the font size of the compound numbers in charts 1 and 2.

The font size of the compound numbers in charts 1 and 2 has been increased.

  1. Lactam ring numbering.

As in Charts 1, 2 and 3, ring numbering for the lactam core is indicated in Figure 2.

  1. Line 241: authors’ write; …remarkably, 4g…. However, in terms of cytotoxicity 4g seems to be similar to 4h.

Although the toxicity of 4g and 4h is similar against A549 and SKOV3 cell lines, what we remark in this line is that 4g shows toxicity against RKO cell line, while 4h is inactive.

  1. Line 273: …selectivity with respect to non…

Corrected.

  1. Compound 6c showed improved IC50 value against A549 cells.

Corrected.

  1. Line 372: …good selectivity compared to SKOV3 cell line and healthy cells (MRC5).

Corrected.

  1. line 385: correct the word “significant”. 

Corrected.

Reviewer 2 Report

The document pharmaceuticals-1318653 is well developed in the chemical component and has some flaws in the biological one (example: lack of scales in the microscope images, errors and deviations in the graphics). The informational support with the NMR and HPLC images of all new compounds remains to be placed. There are writing errors because a more careful reading is lacking (CO2, line 363, etc.)
This Reviewer suggests a summary of the manuscript in Molecules (MDPI, with S. Information) and not Pharmaceuticals.

Author Response

The document pharmaceuticals-1318653 is well developed in the chemical component and has some flaws in the biological one (example: lack of scales in the microscope images, errors and deviations in the graphics). The informational support with the NMR and HPLC images of all new compounds remains to be placed. There are writing errors because a more careful reading is lacking (CO2, line 363, etc.)

The scales in the microscope images, errors and deviations in the graphics have been added.

Full experimental details, characterization and 1H, 13C, 19F and 31P-NMR copies and HPLC chromatograms of all compounds, flow cytometric assays on A-549 cells S156 and visualization of cell growth and morphology of A-549 cells is supplied in ESI.

The manuscript has been checked again for errors and typos.

Reviewer 3 Report

In this article the authors evaluate the in vitro cytotoxicity of several highly functionalized ϒ-lactam derivatives against several human tumor cell lines. The studies performed showed that some compounds had in vitro cytotoxicity and selectivity. Authors also gave a SAR and studied the mechanism of action of the active compounds. As the synthesis gives a racemic mixture, the authors synthesized both enantiomers of compound 4v and studied the cytotoxic activity of the racemic substrate and of its two enantiomers.

The manuscript is written in fluent style and there seem to be no spelling mistakes. This contribution may lead to new therapeutic agents in the future. In the article they synthesized and evaluate 48 compounds (although less than half of them have not been synthesized before).

There are some issues which I would like to address:

-Regarding the conditions of the multicomponent reaction used for the synthesis of the lactam derivatives: there is no mention of the optimization of these conditions. Since the reaction seems to be optimized in previous publications by the authors they could comment it in the current manuscript (for a record of why they use those conditions).  

-It would be interesting to address if the compounds comply with Lipinski's rule of five (their physicochemical properties) and their ADME-Tox properties.

-Why was compound 4b selected for the initial testing against seven cancer cell lines? Why were the rest of the ϒ-lactams only tested against some cell lines (RKO, SKOV3, A549 and MRC5) and not the seven of them? Since the selection of these cell lines seems to be based on the activity of compound 4b, and the cytotoxicity was found to be strongly dependent on the substituents, other ϒ-lactams could be active against the rest of the cell lines not selected.

-Authors address the presence of a stereogenic carbon in the lactam derivatives and the possibility of enantiomers having different behaviour. Although they performed the enantioselective synthesis of lactam 4v and tested the enantiomers finding no significant differences between enantiomers and the racemic sample (IC50 values of 9.23±1.68 μM (R), 10.53±0.80 μM (S) and 11.29±1.80 μM for the racemic mixture), it was only tested for one compound and in one cell line (A549).  

-There are some differences between the yield of compounds reported in the material and methods section and in the supporting information with the ones shown in the charts. Like 4a, 5c and 5d from chart 1; and 8b and 11a from chart 2. In most cases the difference is not very big, but in compound 5c it goes from 63% (chart 1) to 11%.

-In material and methods, when giving the description of compounds 19 and 20a there are several CH missing in the 13C NMR. Also, authors selected this compounds, whereas in the supporting information they don’t give the characterization of them stating that their “physical and spectroscopic data are in agreement with the literature”. So it would be better if the authors provide in the manuscript the characterization of new derivatives.

-Note that several schemes have some overlapped text making it difficult to understand (schemes 2 and 4).

-References 1 and 2, which refer to the current situation, should be more current.

-Please note that references 59 and 60 couldn’t be found in the text.

-Also, in line 363 there’s an “extra” text ([¡Error! Marcador no definido.]).

-In the Supporting Information:

Please check the 1H NMR of compounds 5c and 11a as there seem to be missing the integration (in 5c the one at 7.09 and in 11a the one at 6.17).

Please check the 13C NMR of compounds 4e, 4m, 7 and 12 as there seems to be several carbon missing.

References 1 and 3 are the same (Org. Lett. 2018, 20, 317-320).

The conditions in the schemes of the synthesis of 16 and 19 are overlapped.

Also, the 1H NMR of compound 4g doesn’t appear (page S25).

Author Response

  1. Regarding the conditions of the multicomponent reaction used for the synthesis of the lactam derivatives: there is no mention of the optimization of these conditions. Since the reaction seems to be optimized in previous publications by the authors they could comment it in the current manuscript (for a record of why they use those conditions).

Some comments on the optimization of stoichiometry, solvent and Brönsted acid have been added in the text.

  1. It would be interesting to address if the compounds comply with Lipinski's rule of five (their physicochemical properties) and their ADME-Tox properties.

Lipinski's rule of five have been calculated and Gastro-Intestinal absorption and Blood-Brain-Barrier permeation have been predicted. The table is present in the Supporting Information.

  1. Why was compound 4b selected for the initial testing against seven cancer cell lines? Why were the rest of the ϒ-lactams only tested against some cell lines (RKO, SKOV3, A549 and MRC5) and not the seven of them? Since the selection of these cell lines seems to be based on the activity of compound 4b, and the cytotoxicity was found to be strongly dependent on the substituents, other ϒ-lactams could be active against the rest of the cell lines not selected.

We normally make an initial screening on several cell lines and then focus our SAR study into the more sensitive cells. The optimal structure is then tested again in several cell lines. However, we did not find any relevant activity for our best compounds in the other cell lines. During our investigations in the last years, we found that A549 cells are often more sensitive than other cell lines, such as SKOV3 or RKO cells.

  1. Authors address the presence of a stereogenic carbon in the lactam derivatives and the possibility of enantiomers having different behaviour. Although they performed the enantioselective synthesis of lactam 4v and tested the enantiomers finding no significant differences between enantiomers and the racemic sample (IC50 values of 9.23±1.68 μM (R), 10.53±0.80 μM (S) and 11.29±1.80 μM for the racemic mixture), it was only tested for one compound and in one cell line (A549).

The preparation of enantiopure samples implies complex experimental procedures with chiral catalyst and tedious purification methods with several crystallization steps. In order to isolate one pure enantiomer, it is very important, if not mandatory, that the molecule allows the obtaining of crystals, which is not the case of most of our substrates. However, the experiment on both enantiopure samples of one of the compounds concludes that the origin of the citotoxicity is not enantiomer-sensitive.

  1. There are some differences between the yield of compounds reported in the material and methods section and in the supporting information with the ones shown in the charts. Like 4a, 5c and 5d from chart 1; and 8b and 11a from chart 2. In most cases the difference is not very big, but in compound 5c it goes from 63% (chart 1) to 11%.

We thank referee Nr.2 for his careful revision. The yields of all compounds have been checked and corrected.

  1. In material and methods, when giving the description of compounds 19 and 20a there are several CH missing in the 13C NMR. Also, authors selected this compounds, whereas in the supporting information they don’t give the characterization of them stating that their “physical and spectroscopic data are in agreement with the literature”. So it would be better if the authors provide in the manuscript the characterization of new derivatives.

Only the characterization of new compounds is now provided in the manuscript.

  1. Note that several schemes have some overlapped text making it difficult to understand (schemes 2 and 4).

Schemes 1,2 and 4 have been reformatted. Structures in scheme 4 have been rearranged in order to fit in the page at the maximum size. Figure 1 and 2 and Scheme 3 have been also modified.

Schemes have

  1. References 1 and 2, which refer to the current situation, should be more current.

Reference 1 has been updated to 2020 statistics. However, although reviews on chemotherapy in specific cancer types can be found in 2021, there is not a general review on surgery and chemotherapy more current than ref 2 (2018).

  1. Please note that references 59 and 60 couldn’t be found in the text.

Corrected and inserted in the right point.

  1. Also, in line 363 there’s an “extra” text ([¡Error! Marcador no definido.]).

That was reference 59. Corrected.

  1. In the Supporting Information:
  • Please check the 1H NMR of compounds 5c and 11a as there seem to be missing the integration (in 5c the one at 7.09 and in 11a the one at 6.17).
  • Please check the 13C NMR of compounds 4e, 4m, 7 and 12 as there seems to be several carbon missing.
  • References 1 and 3 are the same ( Lett. 2018, 20, 317-320).
  • The conditions in the schemes of the synthesis of 16 and 19 are overlapped.
  • Also, the 1H NMR of compound 4g doesn’t appear (page S25)

We thank again referee Nr.2 for his careful revision. 1H and 13C NMR peaks remarked, have been checked and corrected. Some other mistakes, detected in the process, have been also corrected.

Reference 3 has been checked and corrected.

The overlapping in the schemes and the disappearance of 1H NMR image of 4g seems to have happened during the processing of the word file to pdf. We have checked ESI pdf file prior to resubmission.

Round 2

Reviewer 2 Report

Accept in present form.